# Design of High-Performance and General-Purpose Satellite Management Unit Based on Rad-Hard Multi-Core SoC and Linux

**Lu Li** [1,2]**, Junwang He** [1,3]**, Dongxiao Xu** [1]**, Wen Chen** [1,2,*]**, Jinpei Yu** [1,2] **and Huawang Li** [1,2]

[1] Innovation Academy for Microsatellites of Chinese Academy of Sciences, Shanghai 201306, China
[2] University of Chinese Academy of Sciences, Beijing 100039, China
[3] School of Computer Science, Northwestern Polytechnical University, Xi'an 710072, China
[*] Correspondence: wenchen@ustc.edu

**Abstract:** Since deep space exploration tasks, such as space gravitational wave detection, put forward increasingly higher requirements for the satellite platform, the scale and complexity of the satellite management unit (SMU) software are also increasing, and the trend of intelligentization is showing. It is difficult for the traditional SMU based on single-core system on chip (SoC) to meet the various requirements brought by the above trends. This paper presents a high-performance general-purpose SMU design. Based on rad-hard multi-core SoC, we configure and tailor Linux, and design an SMU software architecture with three modes. It has the characteristics of high performance, high reliability, general purpose and scalability, which can meet the needs of the SMU of future complex satellites. Finally, through the application experiment in the background of the space gravitational wave detection project, the performance and application prospect of our proposed SMU are demonstrated.

**Keywords:** satellite management unit; multi-core SoC; Linux; software architecture; space gravitational wave detection





## 1. Introduction

Satellite management unit (SMU) is one of the main service systems to ensure the completion of the mission of the satellite. It consists of the hardware platform, platform software (operating system, board support package, etc.) and application software. The SMU is responsible for the on-board device management, attitude and orbit control, autonomous mission management, telemetry, telecommand and other functions. It is the key to realize the autonomous flight and management of the satellite and improve the reliability and security of the satellite.

In recent years, space scientific exploration has gradually been moving toward deep space, and the autonomy of space mission management requirement is rising. The scale and complexity of SMU software are constantly increasing, and the demand for deploying artificial intelligent algorithms on spacecraft is also growing, which pose greater challenges to the performance of on-board computers, operating systems, and SMU software architectures.

Taking the space gravitational wave detection as an example, because the gravitational wave detection has high requirements for the control accuracy and stability of the spacecraft, it needs control algorithms with higher complexity and the higher control frequency. At the same time, since the satellite is tens of millions of kilometers away from the Earth during gravitational wave detection, the telecommand delay is high, and the one-way delay is more than 200 s, which requires the intelligent autonomous management ability of the satellite [1].

In terms of the performance of the onboard computer, the performance of the single-core processor is gradually not enough to cope with the increase in the computing amount brought by the above changes, and the single-core processor is restricted by Moore's Law

and the influence of the power wall factor, so it cannot further improve its computing capacity under the condition of limited power. Multi-core processors have more advantages in computing power and energy consumption, and in recent years, rad-hard multi-core processors have begun to emerge. Therefore, the use of a multi-core processor in the design of satellite control and data processing unit is an inevitable choice to improve the processing capacity of the satellite information system [2,3]. To give full play to the performance advantage of the multi-core processor and make it easy to use in the SMU, the support of many aspects, such as an operating system, software ecology and software architecture design, are necessary. These become important problems that need to be solved urgently.

At the operating system level, it is difficult for uC/OS to meet the complex multi-interface communication requirements, while VxWorks is expensive and not open source. The Real-Time Executive for Multiprocessor Systems (RTEMS) is a space qualified operating system that supports multiprocessor, hard real time, portable operating system interface (POSIX) and is freely distributable. However, it is considered a single process with multiple threads, which means a relatively large influence between threads. The decreased popularity in the industry leads to the difficulty in transplanting the latest applications and libraries on the ground to the SMU, which adds obstacles to the intelligentization of the satellites. Compared with the operating systems mentioned above, Linux has advantages, including various system functions, modularity, open source, multi-core processor support and numerous intelligent application libraries [4]. Therefore, Linux is more suitable for high-performance SMU for future complex satellites.

As a general-purpose operating system, although Linux has many superior features mentioned above, its real-time performance is limited in the standard configuration, and its kernel size is huge [5]. Therefore, to apply Linux to spacecraft, it is necessary to properly configure and tailor Linux so as to reduce the size of the kernel and improve the real-time performance. Linux has also been partially applied in traditional satellites, but most of them run on single-core processor with small memory. Therefore, they pay attention to extreme kernel size compression and often need to use a real-time operating system (RTOS) on the main computer, while Linux is only used for payload management, such as STRaND-1 [6] and TacSat-1 [7]. The situation with high-performance multi-core SOCs is quite different, requiring the design of a new Linux application scheme to make full use of the high-performance of multi-core processors and meet the mission requirements of future complex satellites. A balance is needed between kernel size, real-time performance, scalability and the support for intelligent applications.

To get the most out of a multi-core processor and take advantage of the benefits of Linux, traditional SMU software architecture based on a single-core processor is no longer suitable. Some existing research designed the SMU software architecture based on the multi-core processor. They are still based on RTOS, and few of them are based on Linux, which is difficult to meet the requirements of complex satellites in the future. He et al. [8] conducted an analysis of dual-core processors and satellite control and data-processing systems, and proposed an asymmetric multiprocessing (AMP) based architecture. The data acquisition/collection thread runs on the core without floating-point unit (FPU), and the other threads run on the core with FPU. The running sequence of the threads is statically arranged. Jan et al. [9] proposed a symmetric multiprocessing (SMP)-based architecture. Each application has a separate partition to realize the isolation and protection. The scheduling unit is a partition, and the partition has no priority. The partition scheduling scheme is determined in advance and repeats for a fixed period. Ref. [10] designed a fault-tolerant processor architecture based on the tri-core processor, which supports three-mode redundancy at the maximum. It can be configured into three modes. In performance mode, the cores run separately. In normal mode, two cores form a dual-mode lock-step fault-tolerant processor, and the other core works independently. In reliable mode, three cores form three-mode redundancy. Although it is a hardware design, it has reference significance to the design of the SMU software architecture.

A new SMU software architecture is necessary. It should be able to improve the parallelism of the SMU software as much as possible to take advantage of the multi-core processor on the premise of ensuring reliability and scalability. It should also be capable of making full use of the Linux system functions for the scalability of SMU and carrying out the calculation of intelligent autonomous management tasks without affecting the real-time tasks so as to meet the needs for increasing the control complexity and frequency and the satellite intelligent autonomous management of future complex satellites.

The main contributions of this paper are as follows:

- We use a rad-hard multi-core hardware platform to enhance the on-board computing power. Linux is used to manage multi-core resources and provide rich system functions for applications. Through Linux, the environment of SMU software is consistent with that on a personal computer (PC), which is conducive to improve the development process and efficiency. Configuration and tailoring to Linux are implemented considering size, real-time, extensibility and support for intelligent applications.
- Based on the hardware platform and Linux, we design the software architecture with three modes. High-performance and high-reliability mode can be switched flexibly according to operating conditions so as to make full use of multi-core resources in various situations.
- Under the background of gravitational wave detection project, the application of the SMU designed in this paper is analyzed, and a large number of experiments are carried out. Comparing with other methods, our design shows a close multi-core acceleration ratio, and has much better expansibility and generality.

The organization of this article is as follows: Section 2 describes the multi-core SoC used in this article. Section 3 describes the configuration and tailoring of Linux. In Section 4, the SMU software transplantation and the architecture of the SMU software designed for SMP multi-core processor and Linux are described, and the application in the background of space gravitational wave detection project is also analyzed. In Section 5, the performance advantages of our SMU compared with the traditional schemes are demonstrated through the experiments based on the actual SMU software, and the effects of the designed SMU are demonstrated combined with the task requirements in the background of space gravitational wave detection project. Section 6 summarizes this paper.

## 2. High-Performance Multi-Core General-Purpose SMU Hardware Platform

### 2.1. Overview

SPARC V7/V8 [11] is a processor architecture of reduced instruction set and has been widely used in spacecraft avionics. Meanwhile, in the architecture of multi-core processor, SMP [12] is widely used on the ground due to its characteristics of uniform memory access, single physical address space, multi-level cache supporting data locality and low communication delay. In recent years, a number of rad-hard SMP SoCs based on SPARC architecture have come forth, such as the GR740 of Gaisler Research [13] and the S698PM of Orbita Aerospace [14], both using the quad-core fault-tolerant LEON4 SPARC V8 processor. Therefore, multi-core SOCs based on SPARC V8 SMP can be used for high-performance SMU.

### 2.2. GR740

The GR740 is the first rad-hard implementation of the European Space Agency (ESA) Next Generation Microprocessor SoC architecture, and is part of the ESA standard microprocessor components roadmap. The simplified architecture diagram of GR740 is shown in Figure 1. In view of the relatively high technical maturity of GR740, this paper uses the GR-CPCI-GR740 development board, which is based on GR740, to verify and test our design of multi-core SoC based SMU.

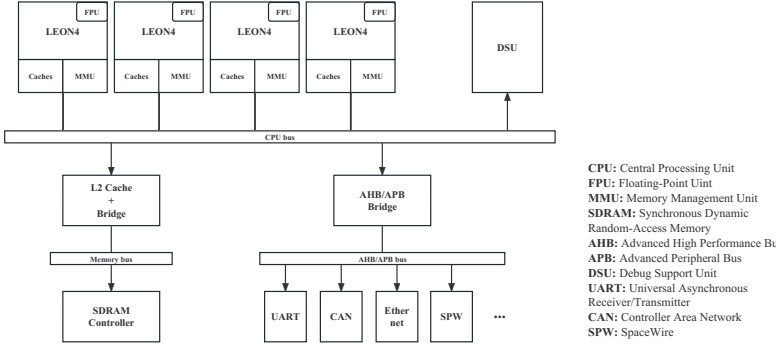

**Figure 1.** Simplified architecture diagram of GR740.

The GR740 is a rad-hard SoC, which has a quad-core fault-tolerant LEON4 SPARC V8 processor with 7 stages, 8 register windows, $4 \times 4$ KiB instruction and 4x4 KiB data caches, memory management unit (MMU), and a double-precision IEEE-754 FPU. The processor runs at a nominal frequency of 250 MHz and is capable of performing one double-precision floating-point operation per core per cycle. The comparison between the main parameters of GR740 [15] and the main parameters of AT697F [16], which is a commonly used onboard processor, is shown in Table 1.

**Table 1.** Main parameters comparison between GR740 and AT697F.

| Parameter | GR740 | AT697F |
|---|---|---|
| Architecture | SPARC V8 LEON4 | SPARC V8 LEON2 |
| Number of Cores | 4 | 1 |
| Processor Bits | 32 | 32 |
| Clock Frequency | 250 MHz | 100 MHz |
| FPU | Yes | Yes |
| MMU | Yes | No |
| Instruction Pipeline | 7 levels | 5 levels |
| Instruction Cache | 16 KB | 32 KB |
| Data Cache | 16 KB | 16 KB |
| Performance | 1700 DMIPS | 86 DMIPS |
| Temp Range | $-40\,°C$ to $125\,°C$ | $-55\,°C$ to $125\,°C$ |
| Typical Power Consumption | 1.2 W | 1 W |
| Total Ionizing Dose (TID) | 300 Krad (Si) | 300 Krad (Si) |
| Single Event Latch-up (SEL) | 125 Mev.cm$^2$/mg | 70 Mev.cm$^2$/mg |
| Single Event Effect (SEE) | <$1 \times 10^{-5}$ events/device/day | <$1 \times 10^{-5}$ events/device/day |

## 3. Operating System Selection and Design

### 3.1. Overview

Because of the high performance of the multi-core processor and the MMU, the GR740 can run Linux directly without using uCLinux.

SpaceX's Falcon 9 rocket demonstrated that Linux with the Preempt-RT patch can be used well in hard real-time systems [17]. For our missions, there are generally two aspects of real-time requirements.

- **Timely data acquisition:** Data need to be fetched from the buffer in time to avoid loss of data. However, interfaces with this requirement, such as the controller area network (CAN), are equipped with direct memory access (DMA) or first in first out (FIFO) buffer to reduce this requirement.
- **Periodic calculation:** The results are used to periodically control attitude, orbit, etc. By adding a handling policy for exceeding deadline, the exceeding deadline probability of less than 5% can be tolerated. If an event-based task has missed its deadline, this task drives no event in this period. If a state-based task has missed its deadline, the state controlled by this task remains the same as the current state.

To sum up, by using Linux with the Preempt-RT patch and enhancing the tolerance of exceeding deadline with small probability in applications, the real time requirements of

our system can be met [18]. Real-time Linux variants, such as RTLinux [19] and RTAI [20], can be avoided.

The benefits of using the mainline Linux kernel are that it makes the environment of SMU software consistent with that on PC, which provides great convenience for building an integrated satellite and terrestrial on-board software development environment. It can use the full Linux system functionality and full hardware performance. Part of the on-board software can be developed and debugged on the desktop Linux computer and easily transplanted to the on-board computer environment. It will lead to great improvement of the efficiency of on-board software development, which is very important for the increasingly complex on-board software [21]. Advanced applications used on the ground Linux system, such as database and Docker, as well as mobile AI libraries, such as Tengine [22] and NCNN [23], can be conveniently transplanted to the SMU software, which is conducive to the development of onboard intelligence. Therefore, the high-performance SMU can adopt mainline Linux with Preempt-RT patch as the operating system.

### 3.2. Linux Kernel

The Linux kernel version used in this article is 5.10. The kernel supports SPARC V8 systems with FPU and MMU in single-core and SMP configurations. The kernel consists of three parts: the Linux mainline kernel, the LEON kernel patch, and the GRLIB driver package. The Linux mainline kernel is a long-term stable version of the official Linux kernel that is actively maintained by the large community and industry. The LEON kernel patch is a LEON support for the Linux kernel actively developed by Cobham Gaisler. The purpose of the GRLIB driver package is to provide support for some intellectual property (IP) core parts of the GRLIB IP library [24]. The structure of the Linux kernel is shown in Figure 2.

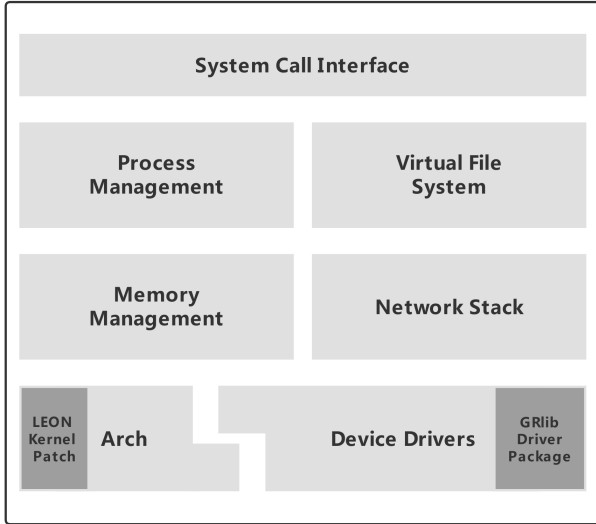

**Figure 2.** Linux kernel structure.

### 3.3. Configuration

In this paper, the configuration file of 5.10 kernel provided by Cobham Gaisler is used as the basis [25] for further configuration and tailoring and is considered the default configuration. Due to high processor performance and sufficient memory capacity, considering the needs of intelligent applications, the basic idea of our configuration and tailoring is not to pursue extreme size tailoring or real-time performance, but to provide support for advanced applications and AI libraries, and to use modularity to achieve scalability. The main contents are as follows:

In terms of real-time performance, the Preempt-RT patch [26] is used to set the preempt model as the fully preemptible kernel. Some of the changes it brings are as follows: converting interrupt service routine to a real-time (RT) thread with priority 50; converting soft interrupt request (softIRQ) to a RT thread with priority 49; turning spinlock into mutex

with priority inheritance; and enabling the high precision timer. Except for very low-level and critical code paths, all of the kernel code is preemptible, making the system more responsive. The timer frequency is set to 1000 Hz to provide fast response.

In terms of memory management, the swap function is disabled because it makes the memory access time difficult to predict and greatly affects the real-time performance of the program. Simple list of blocks (SLOB) is used as the Slab allocator, which is specifically designed for embedded devices with small memory capacity.

For the file system, use random access memory file system (Ramfs) as the root file system and crop out the other file systems. At present, usually there is no file system used on the SMU, and the data are stored sequentially on the non-volatile memory according to the pre-specified area. Ramfs is sufficient to provide the functionality needed to run Linux-based SMU software, including the storage of the C libraries and the software itself. The above approach already satisfies the basic file system requirements of an on-board Linux system. Since Ramfs is compressed in the image along with the Linux kernel and decompressed into memory during system startup, it will also help to speed up the reset of the SMU. If the file system is needed to manage the data on the non-volatile memory for advanced requirements such as on-board database, other file system modules, such as Second Extended Filesystem (EXT2) and New Technology File System (NTFS), can be installed after the system is started due to the advantages of Linux modularity.

For the C libraries, choose Glibc over uClibc. Although uClibc is much smaller in size than Glibc, the memory capacity on a high-performance hardware platform is often more than 128 MB, which is enough to support Glibc. At the same time, because uClibc does not include all Glibc interface implementations, and the development of ground applications and function libraries often use Glibc, using uClibc is not conducive to the transplantation of these functions. While it is convenient to replace the library files in the root file system, the application needs to be recompiled, and there is a risk of incompatibility, so using Glibc directly is more appropriate.

In terms of the network, enable network support, but tailor the drivers for specific interface types. CAN is the interface type widely used in the current onboard computer, and there are more and more attempts to use Ethernet as the payload data interface, which has a certain application prospect. Since both of them depend on the network support function on the Linux system, the network support is enabled. Thanks to the modular design of Linux, the drivers can be installed and uninstalled as modules after the system is running according to requirements, so they can be tailored.

In terms of device drivers, keep only the serial port driver and cut out other device drivers. Since the interaction with the Linux console requires the use of a serial port, the serial port driver is retained. Driver modules of other device drivers can be installed after the system is running according to specific requirements.

For other aspects, due to the need of in-orbit function upgrading of complex satellites in the future, and the need to use different file system modules or device driver modules according to the functions of different satellites, it is necessary to enable the modular function of Linux, and support the installation and uninstallation of modules. Since debugging and test-related functions may affect system performance, especially embedded systems, we disable debugging and test-related functions during the experiments. Since on-board applications are trusted and satellite-ground and inter-satellite communications are encrypted, the system operates in a secure environment, and the security module of Linux can be turned off.

After the above configuration and tailoring, the Linux we use is reduced in size, improved in real-time performance, has scalability, and supports intelligent applications.

## 4. Satellite Management Unit Software Architecture

### 4.1. Overview

Based on the performance of the hardware platform introduced in Section 2 and the function of the operating system described in Section 3, we design a suitable SMU software architecture with the following advantages:

- Three architecture modes are designed to make full use of multi-core processor resources. High-performance and high-reliability mode can be switched flexibly according to operating conditions. Safety design for space application is also considered;
- The SMU software is easy to expand so as to meet the needs of the continuous increase in the scale and complexity of the SMU software;
- A foundation is laid for exploiting the ecology advantages of Linux to provide convenience for the deployment of intelligent algorithms on the satellite;
- The environment of SMU software is consistent with that on PC to improve the development process and efficiency.

### 4.2. Main Modules

The SMU software uses a modular design; each function corresponds to a module and is executed by a thread. The main modules are shown in Table 2.

Each module is equipped with the corresponding fault detection, isolation and recovery (FDIR) techniques.

- **Input and output judgment:** According to the design requirements of each module, the input and output judgment logic of the module is set. When the value cannot pass the judgment, a software module failure occurs. According to the severity of the fault, the fault recovery methods are adopted, including module reset, system soft reset, hardware power-off and reset, or switch to the standby computer.
- **Device monitoring:** The data acquisition module also undertakes the task of device condition monitoring. It periodically collects the characteristic data of each device and analyze it in real time. When the characteristic data of a device exceed their reasonable range or cannot be collected for a certain period of time, the device is regarded as failed. Handle the failure according to the predetermined recovery procedure.

When the computing system cannot work stably, a serious fault occurs, and the system enters safe mode. The safe mode is implemented in the boot software. In this mode, the system has basic telecommand, telemetry and code updating functions, allowing the ground team to troubleshoot and patch the system.

**Table 2.** Main modules of SMU software.

| Module | Abbreviation | Function |
| --- | --- | --- |
| Task Management | TaskMng | Managing the periodic running of the other threads, driven by real-time clock timing |
| Data Acquisition | DAQ | Processing payloads and sensors switch, collecting data from payloads and sensors |
| Orbit Determination | Orbit | Determining the current orbit mode, calculating the orbit data and outputting to other subsystems |
| Attitude and Orbit Control System (AOCS) Determination | AOCS-D | Processing data from star sensors and gyros, determining the attitude and outputting to the attitude control module |
| Attitude and Orbit Control System Control | AOCS-C | Calculating the guidance rate, selecting the attitude control mode, performing the control calculation and outputting the results to the actuators |
| Telecommand Handle | TC | Managing telecommand package storage area, and processing telecommand instructions |
| Telemetry Assemble and Management | TM | Preparing and delivering telemetry channel data |
| Power Management | Power | Managing the power mode according to the battery status |
| Thermal Control | Thermal | Managing thermal control mode, performing closed-loop control based on thermal sensor data and outputting results to the actuators |
| Ground Test | GTest | Handling ground test uplink command, and ground test group package delivery |
| Autonomous Mission Management | AMM | Executing the autonomic management instructions within the time range |

### 4.3. Software Features

In the traditional SMU based on RTOS, the SMU software contains both functional modules and device drivers, and manages the devices. The operating system is mainly responsible for process management and simple memory management. However, SMU based on Linux puts the drivers under the management of Linux, as shown in Figure 3. Using the universal driver interface of Linux, and the design idea that "everything is file", it is conducive to the standardization of the interface of various devices on the satellite, and will greatly improve the development efficiency of the SMU. At the same time, the SMU software and the device drivers are decoupled, which not only reduces the size of the SMU software, but also reduces the development time, and further improves the development efficiency of the SMU.

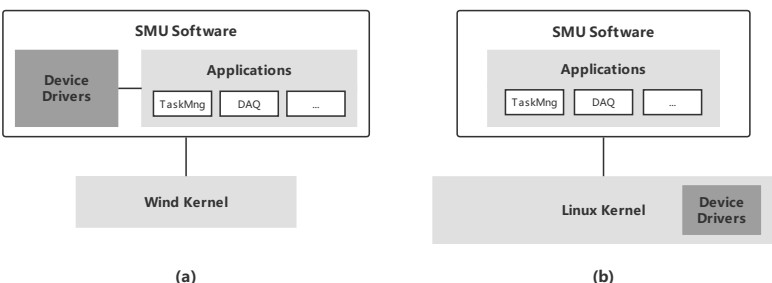

**Figure 3.** Comparison between SMU software architecture based on VxWorks and Linux. (**a**) SMU software architecture based on VxWorks. (**b**) SMU software architecture based on Linux.

Because the traditional rad-hard processor does not have MMU, the SMU software directly uses the physical address when accessing the memory, and often directly operates on the data through the physical address, which has certain security risks. In the SMU based on high-performance multi-core SoC and Linux, because of the MMU, the SMU software uses the virtual address. The SMU software data originally operated through physical addresses are placed into the data segment of the SMU software program, accessed through a virtual address, and isolated from other processes.

Since Linux supports POSIX, the software designed in this paper uses POSIX for thread and semaphore management. Through the POSIX, each thread of the SMU software is set to a real-time thread and other properties that help to improve the real-time performance. To be specific, through the attribution named mlockall, all the current and future memory mapping of the thread is locked to avoid the uncertainty caused by the page fault when the thread reads or writes the memory that has not been committed to the physical memory. Choosing the first in first out scheduler (SCHED_FIFO) or round robin scheduler (SCHED_RR) on demand, with this scheduling strategy, Linux classifies the thread as a real-time thread with a higher scheduling priority. In addition, using POSIX enhances the portability of SMU software in preparation for the SMU software ported to RTOS supporting POSIX and low-performance processors.

### 4.4. Multi-Core Application Mode Design

In addition to the above common features, in order to make full use of the advantages of the multi-core processor under different operating conditions, three different structural modes are designed, including low-power mode, triplication-redundancy mode and high-performance mode. As shown in Figure 4, these three modes can be flexibly and dynamically configured according to the satellite operating conditions and mission requirements.

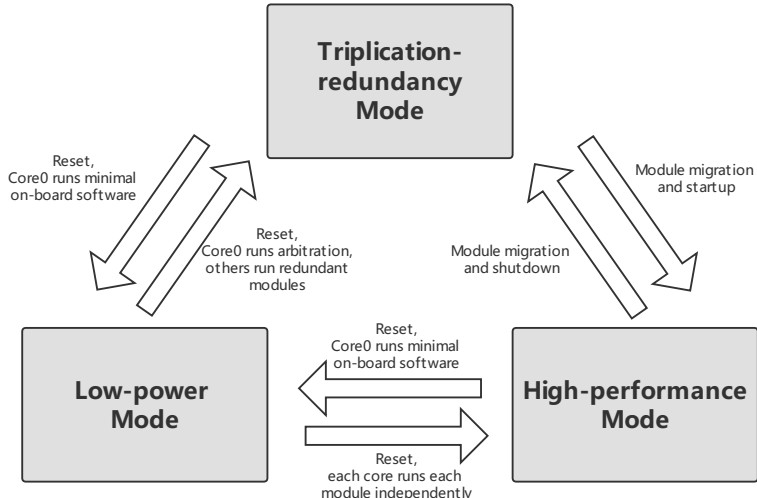

**Figure 4.** Switching between the three modes.

### 4.4.1. Low-Power Mode

This mode is used to reduce the power consumption of multi-core processors by enabling only one processor core and running only basic functions when the space environment is stable and the spacecraft only needs to perform simple tasks, or when the power of the whole satellite is insufficient.

At this time, the SMU software only runs the most basic modules, as shown in Figure 5. In this mode, the SMU software enters the basic operation mode, and each module executes in order of priority, runs on the main core, and the rest cores are shut down.

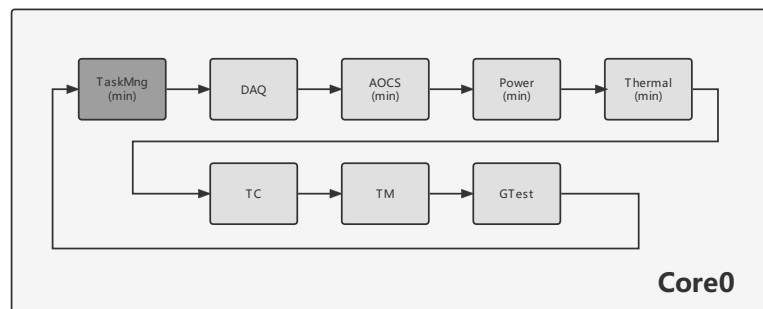

**Figure 5.** Low-power mode.

When entering the low-power mode, the SMU software resets, creates the basic mode TaskMng module on the main core, and closes the rest of the processor cores. The TaskMng initializes the DAQ, AOCS, Energy, Thermal, TC, TM, and GTest in the basic mode. The modules are executed in the same order at each cycle.

### 4.4.2. Triplication-Redundancy Mode

When the satellite operates in the orbit with severe space environment, the probability of the single event effect increases, so it is necessary to switch to the triplication-redundancy mode to increase the reliability of the system.

As shown in Figure 6, the triplication-redundancy mode enables at least four processor cores.

Core0 runs the TaskMng module, which is responsible for controlling and supervising the operation of other modules; it runs the DAQ module and input the collected data of sensors to each fault tolerant core; and it runs the arbitrator module. When each fault tolerant core completes the calculation of a module, the results are broadcast to all cores. The arbitrator module synchronizes and arbitrates the calculation results, and transmits the

arbitrated data to the corresponding actuators. The arbitrator module adopts the voting system. As the instantaneous failure is a small probability event, the final result can be determined by voting. According to the calculation results of the three fault tolerant cores, the result with more votes is taken as the output. Core0 can use the fault tolerance method based on error detection code to monitor and correct its own running state.

Core 1, Core 2 and Core 3 are fault tolerant cores, running the same functional modules respectively. The functional modules on each core are controlled by the TaskMng module running on Core0 and executed in order of priority. After each module completes its task, they wait for Core0 to synchronize (Sync) and arbitrate.

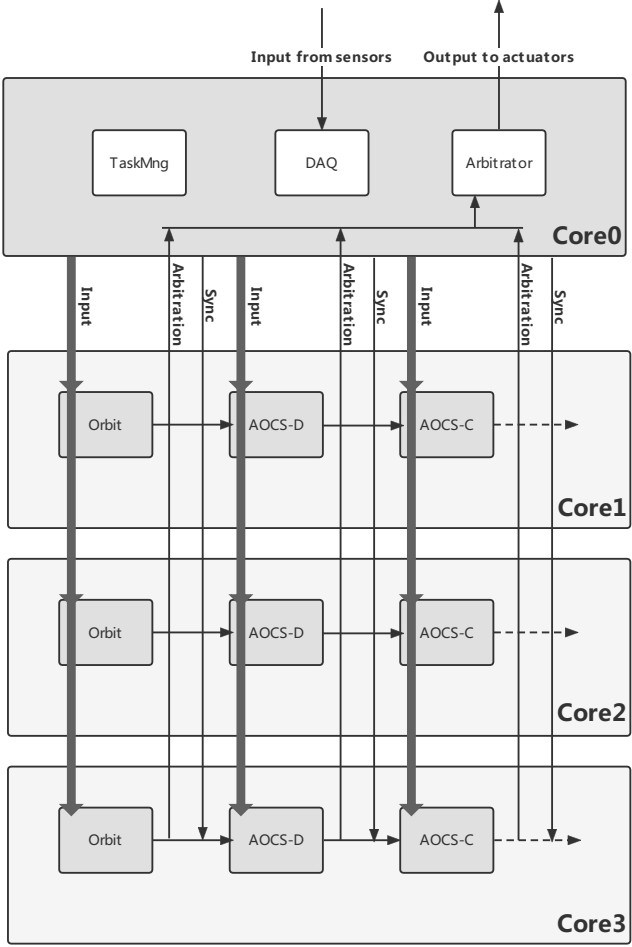

**Figure 6.** Triplication-redundancy mode (Typical).

The above is the typical architecture of this mode. In order to deal with the problem that Core0 acting as a voter becomes a single point of failure, a method of the voter rotation of the cores is further designed. As shown in Figure 7, the relation of cores can be regarded as a directed cyclic graph. When the core as the voter fails, find the next normal node along the cycle to conduct recovery and function exchange. To be specific, if Core0 is the voter and there occurs a failure, the arbitrator module is migrated to Core1. If Core1 cannot pass the arbitration, it means Core1 also fails. Then we migrate the arbitrator module further; if no core can pass the arbitration, the failure is not recoverable and the system needs to be reset. Assume that Core1 is normal; the other functions of Core0 are migrated to Core1. After the recovery of Core0, the original functions of Core1 are migrated to Core0. The role exchange between Core0 and Core1 is complete, and Core1 is the new voter.

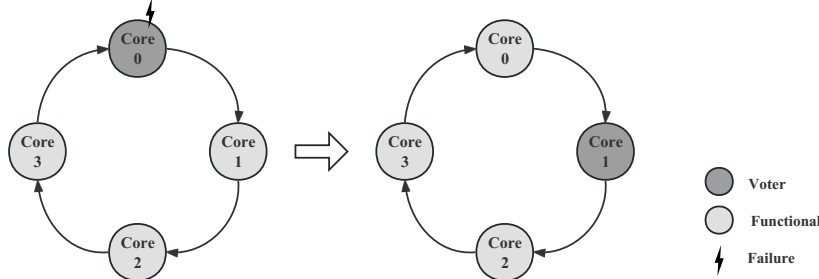

**Figure 7.** Voter rotation of cores.

When switching to the triplication-redundancy mode from the low-power mode, the SMU software resets, and Core1, Core2, Core3 are enabled. The TaskMng module is created on Core0. The TaskMng module creates the DAQ module and Arbitrator module on Core0, and creates copies of all functional modules on Core1, Core2 and Core3 respectively.

When switching to the triplication-redundancy mode from the high-performance mode, the Arbitrator module is created on Core0, and the modules and functions that only run in the high-performance mode are closed. Migrate all remaining functional modules to Core1. Create functional module copies on Core2 and Core3, respectively.

### 4.4.3. High-Performance Mode

SMU software can switch to high-performance mode when the satellite is performing complex tasks that require increased computing power. High-performance mode enables some or all processor cores based on computing power requirements. The modules are distributed to each core as described below, and each core executes them separately so as to increase processing power through parallel computing.

Parallel computing leads to mutual exclusion of data access and dependency between modules. Therefore, it is necessary to change the traditional sequential execution of SMU software into sequential + conditional/time driven combined execution. To this end, the dependencies between modules are analyzed.

- **TaskMng:** Since this module manages the execution of all other modules, this module is not dependent on any other modules and is only driven by real-time clock timing to achieve cyclic execution.
- **DAQ:** This module only relies on the cyclic driven signal of TaskMng.
- **Orbit:** This module is not sensitive to the freshness of data and can use the data obtained from DAQ in the previous cycle, so it only relies on TaskMng.
- **AOCS-D:** The calculation of this module requires the collected data of related sensors and the orbit data in this cycle, so it depends on DAQ and Orbit.
- **AOCS-C:** The calculation of this module requires the attitude data of this cycle, so it depends on AOCS-D.
- **Thermal:** This module is not sensitive to the freshness of data and can use the data obtained from DAQ in the previous cycle, so it only relies on TaskMng.
- **Power:** This module is not sensitive to the freshness of data and can use the data obtained from DAQ in the previous cycle, so it only relies on TaskMng.
- **AMM:** This module is not sensitive to data freshness, and can use orbit data, attitude data, and power mode of the previous cycle, so it only relies on TaskMng.
- **TM:** This module is not sensitive to the freshness of data. All modules have data transferred to this module and the data of the last cycle can be used, so it only depends on TaskMng.
- **TC:** The data of this module are independent of DAQ, so it only depends on TaskMng.
- **GTest:** This module needs data from TM of the current cycle, so it depends on TM.

The relationship that requires data generated by other modules in the current cycle is called strong data dependency, and the relationship that requires data generated by other modules in the previous cycle is called weak data dependency. Based on the above analysis

results, the module data dependency diagram shown in Figure 8 is obtained, according to which the conditions and time-driven dependency of each module are described so as to control the sequential execution of each module.

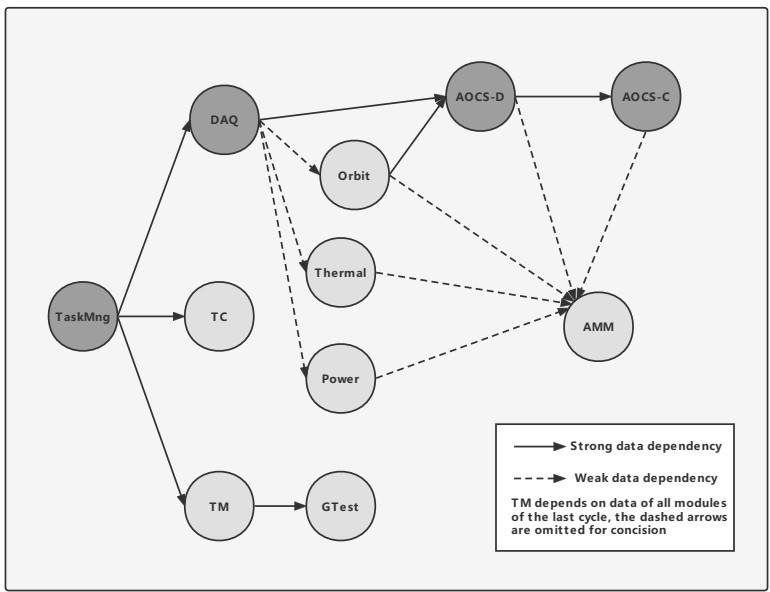

**Figure 8.** Module data dependency.

Since many modules only use the data of the last cycle, the ping-pong data buffer is used to store the data of the last cycle and the current cycle, respectively. The write operation occurs only in the data buffer of the current period. Therefore, the operation of each module reading the data of the previous cycle is not mutually exclusive. The reading of the current cycle data is restricted by the module data dependency diagram and only occurs after the write operation, thus solving the mutual exclusion problem of data access.

According to the module data dependency diagram and the module's different requirements for real-time performance, the module partitioning scheduling scheme is further designed, as shown in Figure 9.

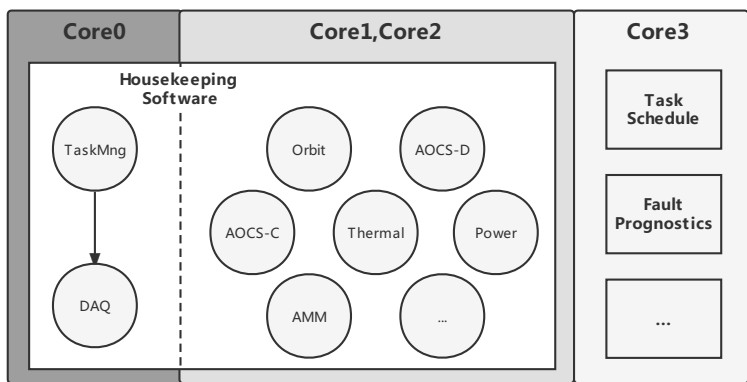

**Figure 9.** High-performance mode.

Because TaskMng and DAQ need a timely response and are sensitive to jitter, these two modules have higher requirements for real-time performance, and are allocated to Core0, isolated from other modules, awakened by periodic clock signals and scheduled with SCHED_FIFO of Linux.

The other modules just need to make sure they meet their deadlines, so they are allocated to Core1 and Core2. Considering the requirements of the subsequent dynamic increase and decrease in modules, and in order to enhance the generality, the scheme of

designing a static scheduling table based on the analysis of the running time of each module is not adopted. Instead, the modules are directly allocated to run on appropriate cores through the multi-core load balancing scheduling of Linux. Each module is scheduled with SCHED_RR of Linux, and the modules are run with dependencies met, not priorities.

Other cores, such as Core3, can be used to run the housekeeping software modules to reduce the load of Core1 and Core2 according to the computing power demand. It can also be isolated from the housekeeping software modules to run intelligent applications independently, such as intelligent scheduling, fault prognostics, etc., to enhance the intelligent autonomous management capability of the satellite.

The above partition scheduling method is implemented using Linux's own system calls and schedulers, which can be expected to be reliable [27], because they are open source and maintained by the Linux community, and are widely used in all kinds of devices on the ground, including servers that need to run stably day and night, which means they have been verified through long time execution.

The existing research is mainly based on the method of designing a static scheduling table [8,9]. The characteristics of our scheduling method are compared with the traditional method of static scheduling, as shown in Table 3.

**Table 3.** Feature comparison between our scheduling method and static scheduling.

|  | Static Scheduling | Ours |
|---|---|---|
| Performance | Optimal scheduling performance can be achieved. | Frequent thread switching brings some acceptable overhead. |
| Scalability and Generality | Any slight change could lead to a redesign, and the running time of each module is needed, which is difficult to estimate when constraints between modules are complicated. | Only need to specify the partition of the modules, and adapt to various scenarios. |
| Reliability | Fixed scheduling scheme is not robust to exceptions, such as a module exceeds its estimated running time. | Using Linux's own system calls and schedulers, which are very reliable. |

The process of switching into high-performance mode is described below.

When switching from low-power mode to high-performance mode, SMU software resets and enables cores, such as Core1 and Core2, on demand. TaskMng is created on the main core Core0, and TaskMng creates modules on each core according to the partition scheduling scheme.

When switching from the triplication-redundancy mode to the high-performance mode, TaskMng migrates the modules to the corresponding core based on the partition scheduling scheme, creates modules that run only in high-performance mode on the corresponding core, and enables the functions that run only in high-performance mode.

4.4.4. A Case Study: Space Gravitational Wave Detection Project

In view of the above three different structural modes of the SMU software, it is necessary to demonstrate its applicable scenarios and necessity through a case study.

The Taiji-Program is a space gravitational wave detection project initiated by the Chinese Academy of Sciences. It is planned to consist of three satellites at the top of a huge equilateral triangle with an arm length of about 3 million kilometers to form a gravitational wave detection constellation. The laser interferometry method is used to measure the small variation in the distance between the test masses carried by each satellite caused by the passage of gravitational wave train so as to conduct direct detection of gravitational waves in medium and low frequency bands (0.1 mHz–1.0 Hz), which is of great scientific significance [28].

The following is an analysis of how the SMU software architecture designed in this paper is applied to the main phases of space gravitational wave detection:

- **Launch and early operations phase:** This phase includes launch-related activities to separate the three spacecrafts in their orbit. The space environment is relatively stable

in this phase, and the tasks performed by the spacecraft are relatively simple, so the SMU software only needs to enter the low-power mode.

- **Cruise phase:** This phase includes a period of about 400 days, during which the three spacecrafts leave Earth and enter their respective orbit. During this process, the space environment of the satellites is relatively harsh, and the errors in the calculation will cause large deviations in the orbit, which need to be corrected at considerable cost. At this time, SMU software needs to use the triplication-redundancy mode to improve the reliability of the calculation results.
- **Commissioning phase:** This phase mainly includes drag-free testing and instrument calibration. Drag-free control is one of the key technologies to obtain the super-static and super-stable space experiment platform, which is to offset the non-conservative force of spacecraft by controlling the thrust generated by micro-thrusters. When carrying out gravitational wave detection tasks, drag-free control is needed to protect the test mass from non-conservative force interference and meet the needs of large-scale constellation laser interferometry [29]. Because the drag-free control involves a large number of high-dimensional matrix operations, the computational complexity is high, which puts forward a greater demand on the on-board computing resources. At this time, SMU software needs to use the high-performance mode.
- **Science operations phase:** This phase carries out several years of gravitational wave detection and scientific data acquisition. In the process of detection, besides the drag-free control, the temperature, attitude and orbit of the satellite are also required to be controlled with higher precision. At this time, SMU software needs to use the high-performance mode to meet the huge demand for computing power.
- **Fault handling:** When the satellite failure causes the power shortage of the whole satellite, SMU software needs to use the low-power mode to ensure that the satellite can maintain the most basic functions so as to troubleshoot and restore the normal working state.

Through the above case study, it can be seen that different phases in the process of gravitational wave detection have different requirements for on-board computational resources and computational redundancy capacity, and therefore have different requirements for the architecture of the SMU software. The SMU software designed in this paper can dynamically switch the structural mode with the change of the phase and state so as to better solve this problem.

## 5. Experiment and Verification

In order to demonstrate the performance of the SMU designed in this paper, a large number of experiments were conducted to obtain the performance data and compared with the traditional methods. Then space gravitational wave project was taken as a case study to show the effect of the SMU designed in this paper.

### 5.1. Configuration

Based on the GR-CPCI-GR740 development board and the Linux after configuration as described in Section 3, SMU software in high-performance mode was selected, and the data of various aspects of performance collected during experiments were analyzed.

Due to the complexity of applications, the operating system, and the multi-core processor, accurate static analysis of the worst-case execution time (WCET) is difficult. We adopt a rough mixed analysis technique to measure WCET. Dynamic analysis is performed on the basis of rough static analysis of the applications. By setting various state parameters of satellite in the program according to the rough static analysis on the functional level, the path length of program execution is increased as much as possible. For example, the AOCS-C module is set to the path that involves the calculation of the extended Kalman filter; the TaskMng module is set to conduct forecast and estimation, including the power estimation for the execution of the mission, antenna selection calculation, etc. Further, a large number of dynamic running tests were carried out to measure WCET.

In terms of the operation system, the configured Linux image size is 2.50 MB, 54.4% less than the default size of 5.48 MB. The number of kernel threads after system startup is 41, which is 19.6% less than the default configuration of 51.

In the aspect of SMU software, because the functions in high-performance mode are the most complex and cover the performance parameters required by other modes, SMU software runs in high-performance mode. As the current SMU software has a small demand for computing power, in order to simulate the increase in computing power demand brought by the improvement of the control precision of future satellites, the control frequency is raised from 8 Hz to 10 Hz based on the Space Variable Objects Monitor (SVOM) [30] SMU software. Moreover, the calculation of the extended Kalman filter described in [31] is added into Orbit, AOCS-D, AOCS-C and Thermal. The time consumed and operation cycle of each module are shown in Table 4. Due to the limited number of interfaces on the development board and for ease of testing, the Socat software on the Linux PC was ported to create a virtual interface, and its administrative process was run on Core3 isolated from SMU software modules.

**Table 4.** Time consumed and operation cycle of each module.

|  | Cycle | Avg (us) | Min (us) | Max (us) |
|---|---|---|---|---|
| TaskMng | All | 300.47 | 232 | 823 |
| DAQ | All | 26,805.64 | 23,480 | 35,643 |
| Orbit | All | 8366.52 | 8334 | 8550 |
| AOCS-D | All | 8543.74 | 8488 | 8872 |
| AOCS-C | All | 8356.91 | 8329 | 8431 |
| TC | A | 487.48 | 472 | 504 |
| TM | All | 134.60 | 42 | 878 |
| Power | B | 29.75 | 29 | 35 |
| Thermal | C | 8489.96 | 8394 | 8636 |
| GTest | All | 37.00 | 30 | 128 |
| AMM | G | 33.23 | 33 | 34 |
| SADA | H | 14.23 | 14 | 15 |

## 5.2. Performance Experiment

Firstly, the SMU is connected to the ground test computer through the serial port, and the ground test software is used to send the telecommand instruction to the SMU. By observing the telemetry data received by the ground test system, it can be confirmed that the SMU designed in this paper can work properly. Then, according to the requirements of the SMU, the response time jitter, deadline and multi-core speedup are tested, respectively. Finally, the reliability improvement of the triplication-redundancy mode is also evaluated.

### 5.2.1. Response Time Jitter

Each module of SMU software is a periodic task, TaskMng is awakened by the periodic clock signal, and the periodic execution of the other modules is controlled by TaskMng. Therefore, the jitter of the response time of the periodic clock signal is very important for TaskMng, and reflects the real-time performance of the system. We run the SMU software for 10,000 cycles, record and analyze the jitter of the response time, and compare the results with the default configuration of Linux. The results are shown in Table 5. The comparison of the jitter distribution is shown in Figure 10.

**Table 5.** Comparison of jitter.

|  | Avg (us) | Min (us) | Max (us) | Var |
|---|---|---|---|---|
| Ours | 1.27 | 0 | 30 | 4.15 |
| Default | 5.07 | 0 | 66 | 53.17 |

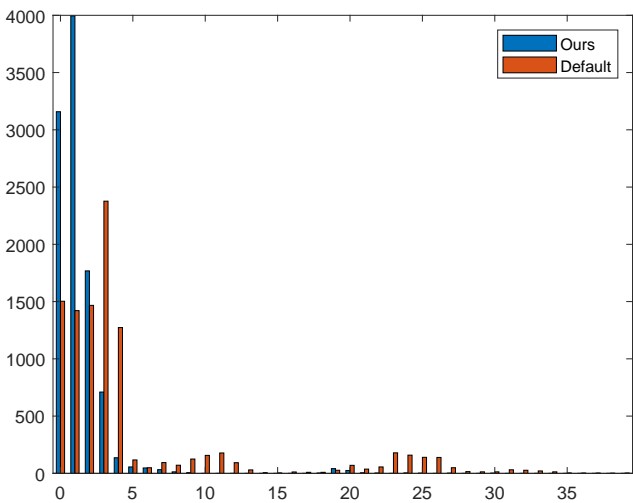

**Figure 10.** The comparison of jitter distribution.

By comparison, under the Linux configuration designed in this paper, the average response jitter of periodic signals is reduced by 3.79 us, the maximum jitter is reduced by 36 us, and the variance of jitter is reduced by 49.02, compared with the default configuration. At the same time, the maximum jitter is less than 0.05% of the cycle length, which meets the requirement of response time jitter for cycle execution of modules.

### 5.2.2. Deadline and Multi-Core Speedup

Since the periodic tasks need to be completed before the deadline, 10,000 cycles are run, the execution time of each cycle is recorded and analyzed, and the results are compared with the results using only one processor core to test the multi-core speedup. Moreover, it is compared with the results of the traditional method of designing the static scheduling table to analyze the impact of the scalability and generality of our proposed scheduling method on the performance. The specific scheduling schemes are compared as shown in Figure 11, and the running time is compared as shown in Table 6.

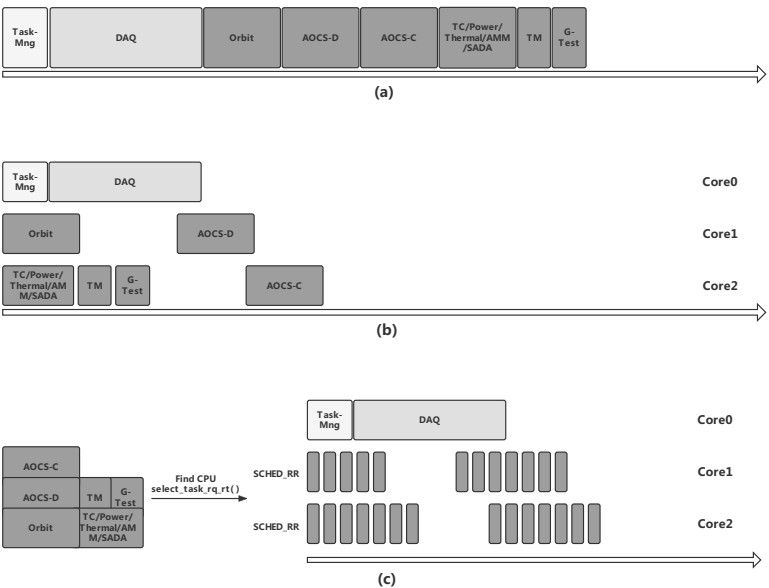

**Figure 11.** Comparison of scheduling schemes for performance experiment. (**a**) Single-core scheduling scheme. (**b**) Static scheduling scheme. (**c**) Our scheduling scheme.

**Table 6.** Running time comparison for performance experiment.

| | Average Running Time (us) | | |
| --- | --- | --- | --- |
| Cycle | Single Core | Static Scheduling | Ours |
| A | 56,283 | 40,453 | 41,428 |
| B | 50,008 | 41,304 | 41,333 |
| C | 59,581 | 42,854 | 42,746 |
| D | 49,923 | 41,608 | 42,045 |
| E | 59,168 | 39,865 | 40,222 |
| H | 51,126 | 42,962 | 43,230 |
| G | 60,402 | 39,814 | 40,048 |
| H | 53,302 | 44,811 | 45,234 |
| I | 49,931 | 41,354 | 41,795 |
| J | 50,022 | 41,528 | 41,917 |

By comparison, it can be seen that our proposed scheduling scheme can meet the deadline requirements under the scenario of high computing power demand, and the average maximum multi-core speedup ratio is 1.51, while the average maximum multi-core speedup ratio of static scheduling is 1.52. Moreover, the analysis shows that the factor that restricts the multi-core speedup is that the critical path composed of attitude and orbit control-related modules is too long, and the improvement of the computing efficiency only through parallel execution at the module level is limited. Therefore, it is necessary to parallelize attitude and orbit control-related modules in the future to further improve the multi-core speedup ratio.

The multi-core speedup ratio of the proposed scheduling method is close to that of static scheduling. However, due to the poor expansibility and generality of static scheduling, the scheduling scheme needs to be redesigned when tasks change, and the design of the scheduling scheme needs the running time of each module per cycle. In the multi-core scenario, there are complex constraints between modules. The running time is highly coupled with the scheduling scheme, which needs to be iteratively optimized to obtain better scheduling performance. At the same time, because the in-orbit operation has a high requirement for reliability, it also needs long-time tests to evaluate the reliability of the scheduling scheme. Therefore, the development and debugging time cost of static scheduling schemes can often reach several weeks, which brings great difficulties to the in-orbit update of code and the extension of in-orbit tasks. In contrast, code updates and task extensions for our scheduling method require only the appropriate partitions to be specified for the module that has changed. Therefore, the proposed scheduling method is more suitable for the needs of future complex satellites due to its strong expandability and generality.

### 5.2.3. Reliability Improvement

Triplication-redundancy mode is selected. Its reliability improvement is evaluated for the following orbits (solar minimum):

- **Geosynchronous Earth orbit (GEO):** 36,000 km, AP-8 min for Radiation Belt Model;
- **Low Earth orbit (LEO):** 700 km, 98.7 inclination, AP-8 min for Radiation Belt Model.

In the above environments, the device error rates per day due to heavy ions for the GR740 flight silicon are estimated with Weibull data presented in [32]. The estimated results are $7.81 \times 10^{-6}$ events/device/day (GEO) and $2.09 \times 10^{-6}$ events/device/day (LEO). Based on the device error rates, functional failure rates of triplication-redundancy mode and configuration without redundancy are estimated and compared. The results are shown in Table 7.

**Table 7.** Comparison of functional failure rates.

| | Functional Failure Rate (Events/Device/Day) | |
|---|---|---|
| **Environment** | **Triplication-Redundancy Mode** | **Configuration without Redundancy** |
| GEO | $2.29 \times 10^{-11}$ | $7.81 \times 10^{-6}$ |
| LEO | $1.64 \times 10^{-12}$ | $2.09 \times 10^{-6}$ |

According to the comparison, the reliability in the GEO orbit environment is improved by 5 orders of magnitudes, and the reliability in the LEO orbit environment is improved by 6 orders of magnitudes. It can be seen that the triplication-redundancy mode designed in this paper has a good reliability improvement capacity.

### 5.3. Application Experiment

Based on the background of the space gravitational wave detection project, the application experiment is carried out. According to the requirements of the space gravitational wave detection project and in order to demonstrate the generality and scalability of the SMU, the Ethernet driver is installed in Linux on the basis of the configuration in Section 5.1, and the drag-free control module is added to the SMU software. The running time is shown in Table 8. Furthermore, the data transmission window planning software based on depth-first search designed for the data protection period in the space gravitational wave detection project is added in Core3 as an example of the satellite intelligent autonomous management program.

**Table 8.** The running time of drag-free module.

| | **Cycle** | **Avg (us)** | **Min (us)** | **Max (us)** |
|---|---|---|---|---|
| Drag-Free | All | 18,790.63 | 18,740 | 18,958 |

The SMU is connected to the National Instruments (NI) platform through an Ethernet port, and the semi-physical simulation test of drag-free control is carried out. According to the test results, the control accuracy is kept within the target continuously. At the same time, the average calculation time of the data transmission window planning is 147 s, which can also output the results within an acceptable time, and does not interfere with the execution of real-time modules.

Further, the running time of each cycle after the addition of the drag-free module is analyzed. The specific scheduling scheme comparison is shown in Figure 12, and the comparison result of the running time is shown in Table 9.

**Table 9.** Running time comparison for application experiment.

| | Average Running Time (us) | | |
|---|---|---|---|
| **Cycle** | **Single Core** | **Static Scheduling** | **Ours** |
| A | 78,813 | 40,110 | 41,316 |
| B | 69,003 | 41,079 | 43,120 |
| C | 78,396 | 42,758 | 44,175 |
| D | 68,809 | 41,095 | 43,179 |
| E | 77,786 | 39,954 | 41,140 |
| H | 69,788 | 42,953 | 44,359 |
| G | 78,933 | 39,975 | 41,282 |
| H | 71,893 | 44,986 | 46,145 |
| I | 68,736 | 41,040 | 43,117 |
| J | 68,795 | 41,114 | 43,170 |

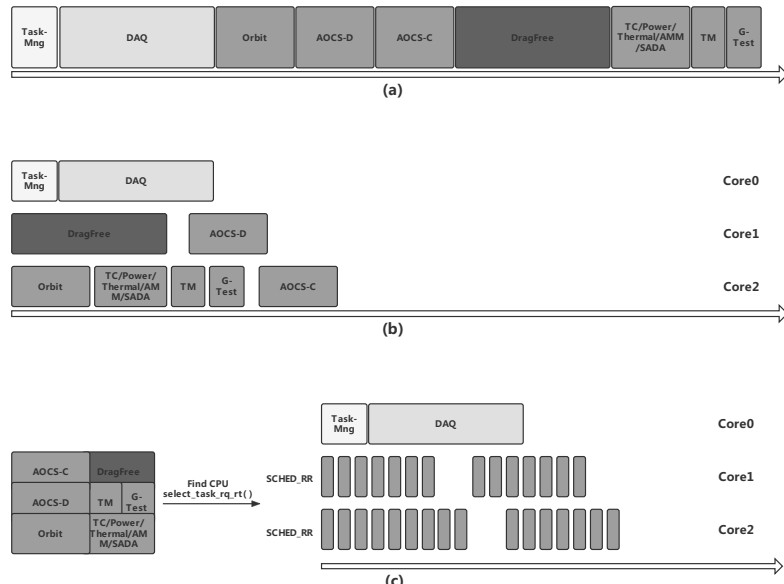

**Figure 12.** Comparison of scheduling schemes for application experiment. (**a**) Single-core scheduling scheme. (**b**) Static scheduling scheme. (**c**) Our scheduling scheme.

By comparison, it can be seen that with the addition of the drag-free module, the average single-core running time of many cycles is close to 80 ms, with the risk of exceeding the deadline, which also shows the necessity of carrying out the design of SMU based on multi-core processor. The average maximum multi-core speedup ratio of our scheduling method is 1.91, while the average maximum multi-core speedup ratio of the static scheduling is 1.97. It indicates that with the increase in the work load, the gap between the multi-core speedup ratio of the two scheduling methods also increases gradually, but still in a small range. The static scheduling scheme needs to be redesigned carefully. Our scheduling method only needs to set the partition of the new module, which is more expandable and general. At the same time, the two methods have no significant increase in the running time after the addition of the drag-free module, indicating that both schemes can make full use of the idle time of the processor caused by the attitude and orbit control critical path and improve the multi-core speedup ratio. The comparison of three methods is summarized as shown in Table 10.

**Table 10.** Comparison of advantages and disadvantages of each method.

|  | Advantages | Disadvantages |
|---|---|---|
| Single Core | Easy to use | Insufficient in performance |
| Static Scheduling | Optimal performance | Lack of expansibility and generality |
| Ours | Good expansibility and generality, high performance | Loss of some performance due to frequent thread switching |

It can be seen that the SMU designed in this paper can meet various requirements of future complex satellites, such as space gravitational wave detection satellites due to the increase in the complexity of the control algorithm, the increase in control frequency and the intelligent autonomous management of satellites.

## 6. Conclusions

This paper presents a design of the high-performance general-purpose SMU. Rad-hard multi-core SoC is used to improve the on-board computing power. Linux is used to manage multi-core computing resources and make the environment of SMU software consistent with that on PC. The configuration and cutting of Linux are implemented in terms of size, real-time, extensibility and support for intelligent applications. The SMU software

architecture with three modes is designed to make full use of multi-core processor resources under various situations. High-performance and high-reliability mode can be switched flexibly according to the operating conditions. The software is designed to be expandable and general. Safety design for space application is also considered. In this way, the need for computing power and software management brought by the increase in the scale and complexity of SMU software and the intelligent autonomous management are satisfied.

The performance experiment shows that the average jitter of the response to the periodic signal is reduced by 3.79 us, and the maximum jitter is reduced by 36 us compared with the default configuration, making the maximum jitter less than 0.05% of the cycle length, indicating a significant improvement in real-time performance. Meanwhile, under the task configuration of the performance experiment, the multi-core speedup ratio of the proposed scheduling method is 1.51, which is only 0.01 less than that of static scheduling, but it has higher scalability and generality. Then, based on the low device error rate due to heavy ions of GR740, the triplication-redundancy mode is estimated to make further improvement of the reliability in a GEO orbit environment by five orders of magnitudes. Based on the background of the space gravitational wave detection project, the application experiment and the semi-physical simulation of drag-free control further verify that the SMU designed in this paper can meet the needs of future complex satellites.

Future work includes applying the high-performance general-purpose SMU described in this paper to the new-technology experiment satellite of our institute, continuously improving the design of the high-performance general-purpose SMU through in-orbit application. Eventually, we expect to achieve more effective support for the various types of spacecrafts to meet the requirements of the autonomous management of an intelligent computing system so as to complete a variety of tasks with greater challenge and scientific value.

**Author Contributions:** Conceptualization, L.L., J.H. and D.X.; methodology, L.L., J.H. and D.X.; software, L.L., J.H. and D.X.; validation, L.L.; writing—original draft preparation, L.L.; writing—review and editing, J.H., W.C. and J.Y.; visualization, L.L.; supervision, W.C. and J.Y.; project administration, J.Y. and H.L. All authors have read and agreed to the published version of the manuscript.

**Funding:** This research received no external funding.

**Institutional Review Board Statement:** Not applicable.

**Informed Consent Statement:** Not applicable.

**Data Availability Statement:** Not applicable.

**Conflicts of Interest:** The authors declare no conflict of interest.

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
