# Peer review of "Design of High-Performance and General-Purpose Satellite Management Unit Based on Rad-Hard Multi-Core SoCand Linux"

_aerospace, doi:10.3390/aerospace10020201_

Round 1

Reviewer 1 Report

This paper presents the design of a satellite management unit (SMU) from both hardware and software points of view. The hardware is based on the GR-CPCI-GR740 multicore development board and the software is based on the Linux kernel with the Preempt-RT real-time patch. The software architecture operates in three different modes, high performance, high reliability and general purpose. The intended application is for a satellite to detect gravitational waves.

The paper is well written and the structure is correct. However, some sentences are too long.

In the introduction, three different operating systems are proposed as elements to support the application software. These are uC/OS, VxWorks and Linux. The first one, uC/OS, is discarded because it does not provide the required support for communications, the second one VxWorks is not considered because of its high price and finally Linux is chosen because of all the support and functionality provided by the community. However, Linux is not qualified for space when there are other alternatives such as RTEMS that are space qualified and are not considered even though they have multiprocessor support. In addition, RTEMS supports hard real time, POSIX interface and is freely distributable like Linux. It is hard to understand that an alternative like RTEMS has not been considered when it has been tested in multiple ESA and NASA missions and is recommended by both agencies.

Although Section 3 states that Linux with the Preempt-RT patch can be considered to have hard real-time support, the reality is that this is not the case. The complexity of a GPOS such as Linux leads to the practical impossibility of executing the static analyses typically performed in RTOSs. This is due to the unrealistic effort involved in calculating the worst-case execution time (WCET) for all control paths and taking into account all possible sources of interference. The common trend is not to consider Linux PREEMPT_RT as a hard real-time solution. Several works such as (Konstantinos Chalas. 2015. Evaluation of Real-time Operating Systems for FGC Controls) or (Jeremy H. Brown and Brad Martin. 2010. How fast is fast enough? Choosing between Xenomai and Linux for real-time applications) lead to the conclusion that real-time Linux systems can only be considered 95% hard real-time. These studies compared the latencies resulting from the execution of specific applications under certain system configurations. The general idea of a 95% hard real-time system is that missed deadlines are tolerated if they occur with a probability of less than 5%. It is not clear from the paper what kind of real time requirements are necessary for the stated mission.

In addition to the three working modes described in section 4.3, a safe mode from which the system can be patched in case of serious failures is missing. Typically this mode is implemented in the boot software prior to loading the operating system and application software.

Section 4.3.2 presents the modular triple redundancy of the high reliability mode. No statistical analysis is presented to determine the improvement of this working option over the configuration without redundancy. In addition, Core0 acting as a voter becomes a single point of failure requiring extensive analysis. No figures are provided regarding reliability improvement.

Section 5 discusses the experiment and verification. The analysis of the worst-case response time is performed by exclusively statistical methods by repeating the tests. It would be interesting to incorporate some static, dynamic or mixed analysis technique that would bring more precision to the worst case execution time (WCET). 

Finally, it would be interesting to discuss in some section the fault detection, isolation and recovery (FDIR) techniques used in the design, verification and validation of flight software.

Reviewer 2 Report

This paper presents a high-performance general-purpose SMU design. Based on a rad-hard multi-core SoC, configuring and adapting Linux, and designing a software SMU architecture with three modes. It has the features of high performance, high reliability, general purpose and scalability, which can meet the SMU needs of future complex satellites. The authors, through the application experiment in the context of the space gravitational wave detection project, demonstrate the performance and application prospects of their proposed SMU. The article seems to be interesting especially for the experimental part, however, the novelty and contribution is not clear, the authors are recommended to check the following points before publishing the article:

1. A more exhaustive review of the state of the art should be carried out, are there other proposals of similar architectures or how does this one differ from the previous ones?

2. Highlight the contributions of the article. The introduction should list the main contributions of the article.

3. Explain in more detail the proposed architecture, as it is one of the important cores of this article.

4. The experimentation part does not compare in terms of advantages and disadvantages with other proposals. 

5. Check that all abbreviations are defined. 

Round 2

Reviewer 1 Report

Most of the previous comments have been addressed in the new version of the paper. The current version is OK to be published.

Reviewer 2 Report

All suggested changes have been addressed. No further comments from me.